# Comparative Analysis of Slip Resistance Test Methods for Granite Floors

**DOI:** 10.3390/ma14051108

**Published:** 2021-02-27

**Authors:** Ewa Sudoł, Ewa Szewczak, Marcin Małek

**Affiliations:** 1Construction Materials Engineering Department, Instytut Techniki Budowlanej, 00-611 Warszawa, Poland; 2Group of Testing Laboratories, Instytut Techniki Budowlanej, 00-611 Warszawa, Poland; e.szewczak@itb.pl; 3Faculty of Civil Engineering and Geodesy, Military University of Technology in Warsaw, 01-476 Warsaw, Poland; marcin.malek@wat.edu.pl

**Keywords:** slip resistance, granite floor, slip resistance value, ramp test, acceptance angle, sliding friction coefficient, comparability of test methods

## Abstract

This paper attempts to compare three methods of testing floor slip resistance and the resulting classifications. Polished, flamed, brushed, and grained granite slabs were tested. The acceptance angle values (α_ob_) obtained through the shod ramp test, slip resistance value (SRV), and sliding friction coefficient (μ) were compared in terms of the correlation between the series, the precision of each method, and the classification results assigned to each of the three obtained indices. It was found that the evaluation of a product for slip resistance was strongly related to the test method used and the resulting classification method. This influence was particularly pronounced for low roughness slabs. This would result in risks associated with inadequate assessments, which could affect the safe use of buildings facilities.

## 1. Introduction

Slip resistance of granite floors is a performance that determines the fulfilment of basic requirement no. 4 (safety and accessibility in use), which, according to Annex I to Regulation (EU) No 305/2011 of the European Parliament and of the Council (CPR) [1], is one of the seven basic requirements to be met by construction works as a whole and by their separate parts. The construction works must be designed and built in such a way that they do not present unacceptable risks of accidents or damage in service or in operation, such as slipping.

Individual European countries have defined more or less specific requirements in this respect. In the UK, the criterion used is the pendulum test value (PTV) of at least 36 units [2]. In Germany, a classification based on the acceptable angle value has been developed, expressed in classes from R9 to R13 [3]. In Italy, the value of the dynamic coefficient of friction for which an acceptable threshold of more than 0.4 has been established is considered for evaluation [4]. In Poland, the issue of slip resistance of granite floors in rooms intended for permanent human occupation is regulated by technical conditions that should be met by buildings and their location [4]. They indicate that the surface of entrances to buildings, external and internal stairs and ramps, passageways in the building and floors in rooms intended for human occupancy, and garage floors should be made of materials that do not pose a slipping hazard. Detailed evaluation criteria are further provided in the Ministry of Investment and Development’s guide, indicating a PTV of at least 36 units [5].

The inadequate slip resistance of a floor carries a risk of slipping. According to the Polish Central Statistical Office, it is, together with trips and falls, one of the leading causes of injuries [6]. These data correspond with the results of analyses conducted by Kemmlert and Lundholm [7] on behalf of the Swedish Council for Occupational Safety and Health, which indicated that slips, trips, and falls account for 17–35% of accidents. Their consequences can be very serious, and treatment can be lengthy and expensive. The most common result is a sprain or fracture of a limb, but more severe cases, such as concussion, have also been reported. It is estimated that one in five slip and fall accidents result in injuries that cause at least one month period of incapacity for work. The slip resistance of floors can be considered a socioeconomically relevant problem.

Slip resistance of floors is determined by several factors, including the properties of the material from which the floor is made, the conditions of its use, the psychophysical state of the user, and the properties of footwear [8,9]. Statistical data indicates that accidents due to slipping most often occur in the autumn–winter period, mainly in public buildings [6] where, as indicated by the analysis of contemporary architectural trends, large-format stone or ceramic tiles with high gloss and smooth surface dominate (Figure 1).

One of the most popular flooring solutions used in public facilities, granite slabs, was used for this study. Slabs of medium-grained Strzegom granite, light gray in color and obtained from the Strzegom–Sobótka massif deposits located in the Sudetes Foreland block, were used. Strzegom granite slabs have been used for flooring in buildings such as the Peace Palace in The Hague, the Palace of Culture and Science in Warsaw, the 10th-Anniversary Stadium in Warsaw, the Congress Centre in Berlin, underground stations in Warsaw, Berlin, and Vienna, numerous office and commercial buildings, railroad stations, underground passages, as well as boulevards, bridges, markets, and squares [10].

Evaluation of slip resistance is conducted using a variety of test methods. Consequently, it is expressed in different parameters that form the basis for independent classifications. Among the most popular is the classification based on slip resistance value (SRV; Figure 2), the acceptance angle value determined by the ramp test (Figure 3), and the dynamic friction coefficient (μ) (Figure 4).

Classification based on SRV, also referred to as PTV, was developed by the UK Slip Resistance Group [2] and introduced into the Health and Safety Executive guidelines [11]. It shows the risk of slipping depending on the value of SRV (Figure 2). It is assumed that the probability of slipping on a floor with an SRV ≥ 36 is 1:1,000,000, while it increases up to 1:20 with an SRV < 24 [2].

The values of the acceptance angle (α_ob_) determined by the shod ramp test are the basis for determining the slip resistance class. There are five classes in accordance with DIN 51130 [3]. Solutions classified as R9 have the lowest slip resistance, while those corresponding to class R13 have the highest (Figure 3).

A separate classification was developed based on the value of the dynamic coefficient of friction (μ) (Figure 4). It is assumed that the floors for which the value of μ is higher than 0.75 can be regarded as antislip [12], while μ above 0.4 is regarded as an acceptable value [4].

The classification scales associated with the various slip resistance test methods are not compatible [13,14,15]. The feelings of people walking on the floors are not always reflected by the values of the coefficients obtained by methods using only test equipment, as shown Choi et al.’s work [14]. An additional source of confusion is the fact that subjective evaluations are expressed on a nominal scale, while measurement results (e.g., sliding friction coefficient (µ)) are expressed on ratio or interval scales.

Attempts are being made to create new classification methods. For example, Çoşkun [16] proposed a new sliding risk scale on natural stones surfaces. Cluster analysis was performed by the author using the *k*-means method. This resulted in a better resolution classification (results were assigned to more classes than in the case of standard methods used). While it would seem appropriate to create a new scale with better resolution that is universal, habits of a particular type of classification and requirements in higher-level documents expressed in the classification scales used in a particular country may prevail in different countries. In view of this, the best way for unification might be to create a matrix of result dependencies and classifications in the existing set, rather than expanding the set.

Following this pattern, this study compares the results attributed to three different classifications. One of the research methods used is a subjective evaluation, but the study results’ obstacle of nominal scale is removed. In the ramp test, the result is expressed in a physical unit of measure (tilt angle expressed in degrees), although its evaluation is typically subjective. The acceptance angle values obtained through the ramp test, slip resistance value, and sliding friction coefficient (μ) are compared in terms of the correlation between the series, the precision of each method, and the classification results assigned to each of the three obtained indices. A full comparison of classifications and the creation of possible conversion factors will be possible after results are obtained for a large group of different materials.

## 2. Materials and Methods

### 2.1. Materials

Granite slabs with different processing textures were used for study, namely polished (PO), flamed (PL), brushed (SZ), and grained (GR). Their characteristics, including roughness parameters, are shown in Table 1.

An Olympus OLS4100 laser scanning digital noncontact microscope (Olympus, Tokyo, Japan) was used to measure roughness parameters. *Sa* and *Ra* values, expressing roughness parameters, were determined according to ISO 3274 [17] and ISO 4288 [18]. The use of the noncontact method (laser beam) in the procedure of measuring the parameters of the geometric structure of the surface, especially the roughness profile, significantly improves the accuracy of the measurement by eliminating the effect of rounding the measuring tip used in the contact method. A 5× objective lens was utilized at 864× total magnification in mixed observation mode. Measurement resolution (laser measurement) was 200 nm. The observed area was 2560–320 µm. The raw *Ra* and *Sa* values were utilized to obtain bulk surface roughness information. *Ra* was measured in 10 different regions, with 10 profiles chosen from each region approximately equidistant from one another. Five equidistant measurements were taken along the sample’s length. The sample was then rotated roughly 180°, and a subsequent five additional measurements were taken. The total number of *Ra* and *Sa* measurements were arithmetically averaged to obtain the final value of *Ra* and *Sa*. To verify the robustness of the method, reproducibility tests were conducted on five samples.

### 2.2. Slip Resistance Value

The SRV test was conducted using an instrument (WESSEX, Aldershot, UK) referred to as a British pendulum (Figure 5). The test technique used was in compliance with EN 14231 [19], which is in accordance with CEN/TS 16165 [20] Annex C. The test was to determine the energy loss of the slider due to friction against the test surface. A Type 57 (CEN) slider made of 55–61 International Rubber Hardness Degrees (IRHD) rubber (WESSEX, Aldershot, UK) was used, with a width of 76.2 mm and a slid length of 126 mm. The frictional force between the slider and the test surface was determined by measuring the pendulum deflection while the slider was moving using the C scale. Before testing, the instrument was calibrated using reference substrates, namely glass, a reference plate, and polishing paper. Measurements were conducted under dry and wet conditions (after wetting both the sample and the slider with distilled water). The test was conducted on two samples of a given solution, with 10 measurements in dry conditions and 10 measurements in wet conditions taken in each series.

### 2.3. Ramp Test

Ramp test was performed using the shoe foot method according to CEN/TS 16165 [20] Annex B, corresponding to DIN 51130 [3]. The test was to determine the acceptance angle (α_ob_), which is the maximum angle of the sample in relation to the level at which a person walking on the floor begins to slip. The researchers walked on ramp (Gabrielli SRL, Florence, Italy) in an upright posture, forward and backward. At the same time, the angle of the sample was changed from a horizontal position (Figure 6a) to an angle at which the researcher no longer felt confident and could not continue walking (Figure 6b). The tests were conducted independently by two researchers. The subjectivity of their experiences was reduced using calibration liners, and the resulting corrections were incorporated into the acceptance angle value. Three standard liners were used for the calibration process. The liners’ acceptance angles were 8.7, 17.3, and 27.3°, respectively. Each person walked on each standard liner three times, and the mean calibration acceptance angle values were determined. Each individual correction value Δ*α* was calculated as a difference between the liners’ acceptance angle and the calibration acceptance angles. Each of the individual correction value Δ*α* was less than the critical differences (≤3.0°). If one of the absolute values was greater, the test person in question would be excluded from the test. Correction value (*Dj*) was calculated from the values obtained from the calibration liners’ values. The calculation of *Dj* was carried out as follows:(1)$$D_{j}=\Delta \alpha -\frac{1}{\sqrt{2}}$$

The test samples were covered with engine oil during testing, and the researchers wore standardized footwear with properly profiled rubber sole. The contact pressure values in the ramp test hovered around the level (1.8–2.0) N/cm^2^ when in static state.

### 2.4. Sliding Friction Coefficient

Sliding friction coefficient (μ) test was performed according to CEN/TS 16165 [20] Annex D. A tribometer (GTE Industrieelelektronik GmbH, Viersen, Germany) equipped with sliders imitating shoe heels, exercising a total contact pressure of 9 ± 1 N/cm^2^ when in static state, was used (Figure 7). Moving along two intersecting paths with a constant speed of 0.2 m/s, the device recorded the frictional force between the slider and the sample. The dynamic coefficient of friction (μ) was calculated as the quotient of the frictional force and the contact force of the slider on the sample. Two different sets of sliders were used in the study. The first set consisted of three sliders with styrene–butadiene rubber (SBR) with a density of 1.23 g/cm^3^ and a Shore D hardness of 50. In the second set, the rear slider was made of SBR and the front sliders were made of tanned leather with a density of 1.0 g/cm^3^ and a Shore D hardness of 60. Prior to testing, the instrument was calibrated with reference substrates, namely glass, high-pressure laminate (HPL), and ceramic tile. Measurements were carried out in dry and wet conditions (after wetting the sample with demineralized water). The test was carried out on two samples of a given type, with 10 measurements in dry conditions and 10 measurements in wet conditions taken for each sample.

## 3. Results

### 3.1. Slip Resistance Value

The SRV obtained in this work is summarized in Figure 8 against the arithmetic mean of the profile deviation from the mean line (*Ra*) expressing the surface roughness. Comparing these results to the criteria developed by the UK Slip Resistance Group [2], it can be concluded that an SRV ≥ 36, which is taken as an indicator of low slip risk (Figure 2), was achieved in dry conditions by all the tested solutions. The SRV depended on the type of treatment texture [21,22]. The lowest SRV value was obtained for PO with *Ra* equal to 1.85 μm (60 units), followed by PL with *Ra* equal to 25.9 μm (77 units) and SZ with *Ra* equal to 27.1 μm (93 units). The highest SRV value was for GR with *Ra* equal to 83.7 μm (98 units). The increase in *Ra* was generally accompanied by a nearly proportional increase in SRV for polished, flamed, and brushed slabs. For the grained slabs, the increase in *Ra* did not fully translate into SRV values. The above may be due to the specific surface profile of the grained slabs. Out of the tested range, only this type of slab has concavities and convexities with a circular shape, which may cause a different adhesion of the slider in the PTV test. As in other works [13,23], a significant reduction in SRV was observed under wet conditions. However, it remained satisfactorily above 36 units for the flamed, brushed, and grained slabs. There was a decrease to 22 units for polished tiles, indicating that the floor poses a very high risk of slipping under these conditions. The SRV results obtained under dry conditions corresponded with the study of Karaca et al. [24], who obtained SRV values ranging from 42 to 74 units for granite slabs with slightly lower roughness than those tested in this study. In contrast, the slip resistance value determined in this study under wet conditions was significantly more favorable. In the aforementioned work, they ranged from 9 to 12 units.

### 3.2. Ramp Test

The above gradation of the slip resistance of granite slabs in terms of SRV values was quite well reflected in the acceptance angle values determined by the shod ramp method (Figure 9). It was found that the least resistant to slipping in terms of α_ob_ values were PO slabs, which according to [3] should be regarded as out-of-class (α_ob_ ≤ 6°), followed by PL of class R10 (10° < α_ob_ ≤ 19°) and GR series of class R12 (27° < α_ob_ ≤ 35°). The highest antislip properties were exhibited by SZ slabs with α_ob_ equal to 34.7°; however, this also places them in class R12. The above classification leads to the conclusion that the resistance of PL series slabs can be considered normal, while SZ and GR series slabs can be considered high (Figure 3), which generally corresponds to the classification according to SRV in dry conditions. However, polished slabs with very low roughness (*Ra* equal to 1.85 μm) are noteworthy. In the SRV test in dry conditions, they obtained a result of ≥ 36, which translates into a high score of slip resistance according to the criteria of the UK Slip Resistance Group [2]. The result of an α_ob_ ≤ 6°, on the other hand, should be considered as disqualifying the solution in the context of floor application.

### 3.3. Sliding Friction Coefficient

The third method used in this study to verify slip resistance was to measure the dynamic friction coefficient, which was conducted under conditions analogous to those used in the slip resistance test. The results obtained with the rubber slider set (μ_R_) are shown in Figure 10 and those with the leather slider set (μ_L_) are shown in Figure 11. In both cases, the dynamic friction coefficient values were close to the value of slip resistance. The lowest μ_R_ and μ_L_ values under both dry and wet conditions were obtained for PO slabs at 0.57 and 0.39 and 0.46 and 0.42, respectively, followed by PL slabs at 0.58 and 0.54 and 0.58 and 0.47, respectively. The μ_R_ and μ_L_ results obtained for brushed and grained slabs were nearly the same level under dry conditions at 0.68 and 0.73 and 0.67 and 0.72, respectively, while the values under wet conditions were 0.66 and 0.60 and 0.67 and 0.59, respectively. Analyzing the obtained μ_R_ and μ_L_ results in the context of the criteria [25] all tested solutions except for the PO series in the test with rubber sliders can be assigned to the level of 0.40 ≤ μ ≤ 0.74, which means that the slabs showed a satisfactory slip resistance in both in dry and wet conditions (Figure 4).

## 4. Discussion

The variety of methods for evaluating slip resistance creates the need to address the relationship between test results obtained by different methods and establish links between classification criteria related to these methods. This issue should be based on testing as many flooring materials as possible.

For measurement methods for which the results are metrologically comparable (i.e., relate to the same physical quantity and are metrologically traceable to the same reference), the issue is relatively straightforward: a method compatibility assessment can be applied to determine whether the differences between results of measurements obtained by different methods are insignificant [23,26].In the present case of slip resistance evaluation, the test methods were determined by the test process’s convention, and their results do not refer to the same physical quantity. The compatibility assessment of the methods in such a case is not justified, and other indirect methods should be used to assess the differences in trueness and precision.

A comparison of the acceptance angle values (α_ob_ using ramp test), SRV, µ_R_ (sliding friction coefficient using rubber slider), and µ_L_ (sliding friction coefficient using leather slider) should also take into account the fact that α_ob_ tests were carried out on a surface covered with engine oil, while SRV and µ tests were carried out on dry surface and on surface wetted with water. Dry surface and wet surface test results differed but were subject to the same classification. Due to the fact that the ramp test was conducted on four surfaces (PO, PL, SZ, and GR) and the other tests were conducted on the same surfaces but with their number doubled using wet surface and dry surface, we used a method of comparison that did not treat the wet and dry surface as separate surfaces but rather considered the tests for wet surface and dry surface as separate test methods.

Pearson’s correlation coefficient within each pair of results was used to compare the results obtained by the seven methods thus defined. The results are presented in Table 2. These were determined for 80 results in each method. All values were greater than the critical value at the confidence level α = 0.05 [27].

The interpretation of the Pearson’s coefficient depends on the purpose and context. For advanced and precise measurement methods, the values shown in the table could be considered insufficient confirmation of the correlation between the results. Each of the methods used in this experiment is characterized by specific arrangements and many noncontrollable factors, namely factors that were not precisely determined in the test model. These factors may cause variabilities taking the form of differences between the results under repeatability and reproducibility conditions.

To compare tests in terms of the dispersion of the results obtained, repeatability standard deviation (s_r_) and reproducibility standard deviation (s_R_) according to ISO 5725-2 [28] are used. However, when the results obtained by different methods are not comparable, standardization of the results would have to be used to obtain information about the differences in precision of the methods. However, classical standardization unifies the dispersion of test results to a standard deviation value of 1. Thus, in this case, quotient transformations where the normalizing values are maximal values for each test method were used. This kind of transformation retains the differences in means and standard deviations [27].

The resulting repeatability standard deviations (s_r_) and reproducibility standard deviation (s_R_) calculated from the results undergoing quotient transformations are shown in Figure 12.

Analysis of the graph shown in Figure 12 indicates that the ramp test method has fairly good repeatability compared to other methods. It defines the angle at which the examiner researcher stopped feeling confident, so the judgment given by the same researcher (expressed by the value of the angle, α_ob_) was not significantly different for the same substrate surface. When another person conducted the research test, their feelings about safety might have varied somewhat, despite their initial mental and physical state. This was confirmed by statistical analysis of the difference between the variance of reproducibility (s_R_) and repeatability (s_r_), which is the component of variance derived from intergroup (interlaboratory) differences:s_L_^2^ = s_R_^2^ − s_r_^2^(2)

The SRV test had a low standard deviation value of repeatability and reproducibility, and the precision of the method depended very little on the surface on which the test was performed. On the other hand, methods based on the determination of the µ factor showed clear differences in precision depending on the surface tested. Table 3 shows the values of the s_L_ deviation.

To determine the statistical significance of differences in the precision of individual methods (ramp test, slip resistance value (for dry and wet surfaces), friction coefficients (µ_R_ for dry and wet surfaces and µ_L_ for dry and wet surfaces), and differences in the precision of testing of individual surfaces (PO, PL, SZ, and GR)), chi-squared test ($\chi^{2}$) was performed for s_L_^2^ variances.

The following criterion was used:(3)$$\chi^{2}=\frac{ns_{L}^{2}}{\sigma_{0}^{2}}<\chi_{\alpha ,n}^{2}$$
where *n* is the number of degrees of freedom, *n* = N − 1 (N = 20 for each value of s_L_^2^). $\sigma_{0}^{2}$ is the variance of the population. As this value is unknown, it was taken as the mean of all s_L_^2^ scores. $\chi_{\alpha ,n}^{2}$ is the critical value for the significance level α = 0.05 and the number of degrees of freedom *n.*

Thus, the $\chi^{2}$ test was used to confirm the hypothesis that the differences of the interlaboratorty component of variance s_L_^2^ of the results obtained for different types of surfaces and different types of tests were not statistically significant.

In Table 3, the values of s_L_ for which s_L_^2^ had not met the condition described by Equation (3) are marked with gray shading. The greatest number of deviating values of interlaboratory variance was found for two surfaces: SZ and GR. This may mean difficulty in ensuring the reproducibility of the results of tests carried out on such surface. As previously established, the SRV method was characterized by the best (as a set for all surfaces) reproducibility.

The precision of the test method, particularly the interlaboratory variance, is extremely important because poor reproducibility of the method increases the risk of incorrect evaluation. From this point of view, on the basis of the presented results, the SRV method should be considered the best (the lowest risk of different assessment by different laboratories).

On the other hand, to get a complete picture of the slip resistance test method, the classification method assigned to it should also be considered, including the classification resolution. 

Different scales of safety risk classifications (Figure 2, Figure 3 and Figure 4) were assigned to the ramp test, slip resistance value, and sliding friction test methods. Figure 13 shows the test results against the classifications assigned to the methods.

The ramp test classification based on acceptance angle values (α_ob_) had the highest resolution. The three surfaces PL, SZ, and GR were assigned to different classes, with the results for PL and SZ surfaces being on the edge of different classes. The PO surface results were below the lower limit of the acceptance angle, indicating an unacceptable risk of slipping.

In the scales assigned to the SRV and sliding friction coefficient tests, PL, SZ, and GR in both wet and dry states as well as the PO floor in dry state were in the same class, i.e., low risk according to the SRV value, while the µ_R_ and µ_L_ values were in the satisfactory class. Only the wet PO floor in the SRV test results reached the high slip risk class. The µ_R_ and µ_L_ values were on the borderline between admissible and satisfactory classes.

The designations for individual classifications were as follows: for ramp test (floor classes): L: low (R9), N: normal (R10), G: good (R11), H: high (R12), and V: very high (R13); for SRV (risk of the slip): Lr: low risk, Mr: medium risk, and Hr: high risk; for sliding friction coefficient (floor classes): d: dangerous, a: admissible, s: satisfactory, and m: model.

The above results indicate that the test method and classification method strongly influenced the slip risk assessment. This supports the thesis regarding the need for uniformity of testing and evaluation rules. Various paths may lead to consistency in assessments, all requiring thorough research. This includes assigning a specific test and classification method to the floor usage conditions; using only one method for all slip resistance assessments; and creating a new research method and a new, consistent classification. A transitional phase could be to provide the possibility of converting results and classifications obtained with different methods to harmonized indices, which could allow comparisons of products in terms of slip risk in situations where slip resistance results are obtained using different methods. This paper is an outline of the problem and an embryo of that phase, showing the relationship between the results and classifications for one type of material with differently prepared surfaces. Creating a matrix to compare slip resistance results would have to involve a large number of test results obtained for different flooring materials. The authors intend to continue to work in this direction.

## 5. Conclusions

This study analyzed the slip resistance of polished, flamed, brushed, and grained granite slabs in terms of slip resistance value, acceptance angle in shod ramp test, and sliding friction coefficients and found that the evaluation of the products in terms of slip resistance was strongly related to the applied testing and classification method. 

The test method’s influence was particularly visible with regard to products with very low roughness, such as polished slabs. These solutions were classified as low slip risk for slip resistance value in dry conditions and satisfactory for dynamic friction coefficient. In the ramp test, they obtained an acceptable angle value at a level that prevented them from being classified in the lowest slip resistance class. 

The test methods were compared in terms of accuracy and resolution of classification. Neither method had been well assessed on both criteria. The SRV method showed the highest precision, resulting in the lowest risk of different classification by different laboratories, but the resolution of the classification assigned to this method was too low. The ramp test method had a resolving classification, but the test method reproducibility appeared to be the worst.

The choice of test and classification method is often dictated by national regulations related to specific applications. However, a question still remains as to which of the assessment methods provides the lowest risk of slippage. According to the authors of this paper, it is not guaranteed by even the most precise method that classifies all surfaces as providing a low risk of slippage. Thus, the safest of the discussed methods of testing and classification, both from the point of view of producer and user risk, seems to be the ramp test.

The testing and classification results indicate an important need to standardize testing methods and classification methods or to create a matrix that allows comparison of results obtained by different methods. 

The authors will continue research on other types of flooring materials.

## Figures and Tables

**Figure 1 materials-14-01108-f001:**
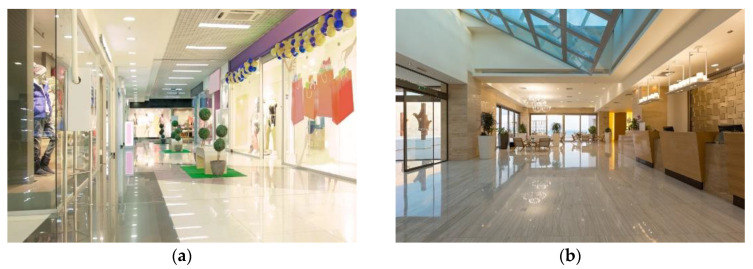
Contemporary floor in public facilities: (**a**) a commercial function, (**b**) a service function.

**Figure 2 materials-14-01108-f002:**
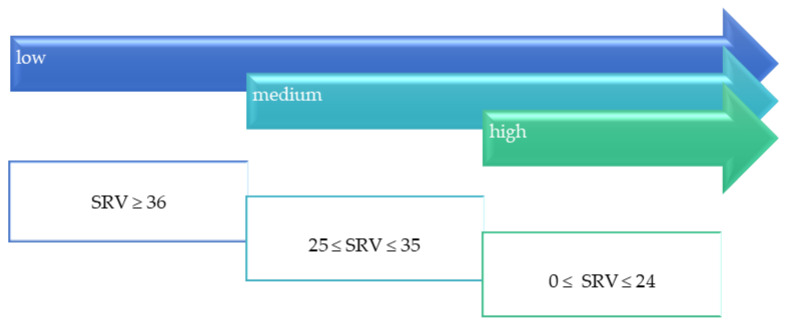
Slip risk in relation to slip resistance value (SRV).

**Figure 3 materials-14-01108-f003:**
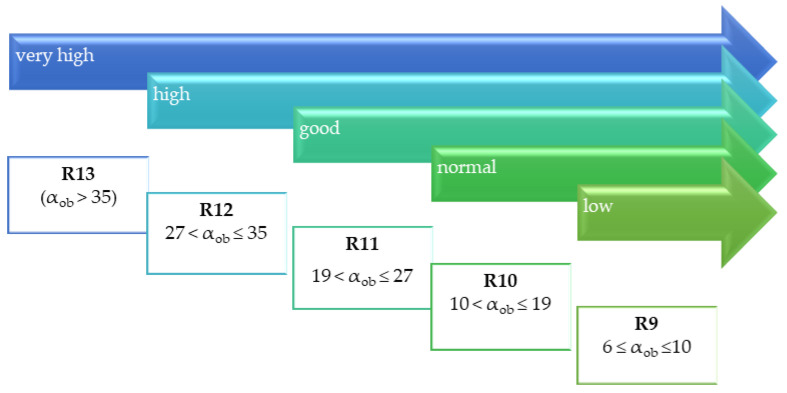
Slip resistance depending on the antislip class and the corresponding values of the acceptance angle (α_ob_) determined by the shoe foot method.

**Figure 4 materials-14-01108-f004:**
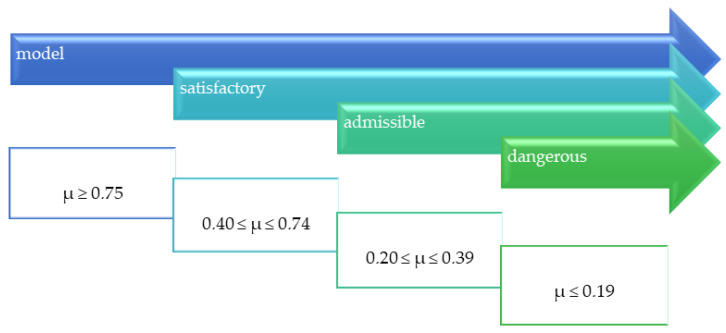
Slip resistance of floors depending on the value of the dynamic coefficient of friction μ.

**Figure 5 materials-14-01108-f005:**
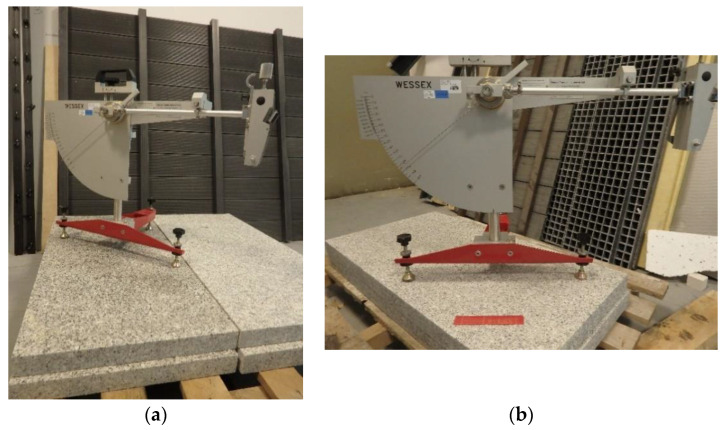
SRV test: (**a**) polished granite slabs (PO) in wet condition, (**b**) brushed granite slabs (SZ) in dry condition.

**Figure 6 materials-14-01108-f006:**
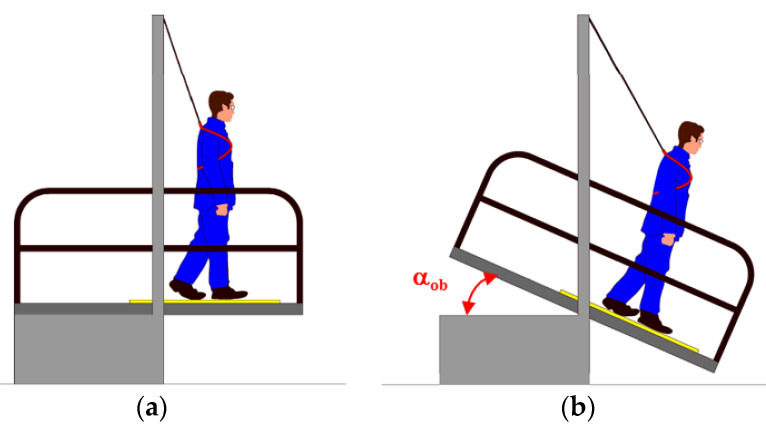
Ramp test: (**a**) starting position, (**b**) position at maximum acceptable angle α_ob._

**Figure 7 materials-14-01108-f007:**
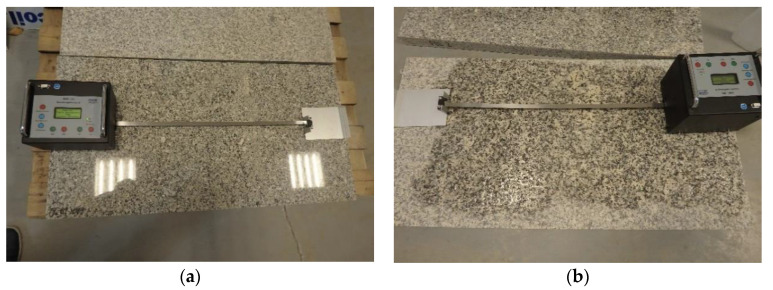
Dynamic coefficient of friction measurement: (**a**) PO series in dry condition, (**b**) flamed (PL) series in wet condition.

**Figure 8 materials-14-01108-f008:**
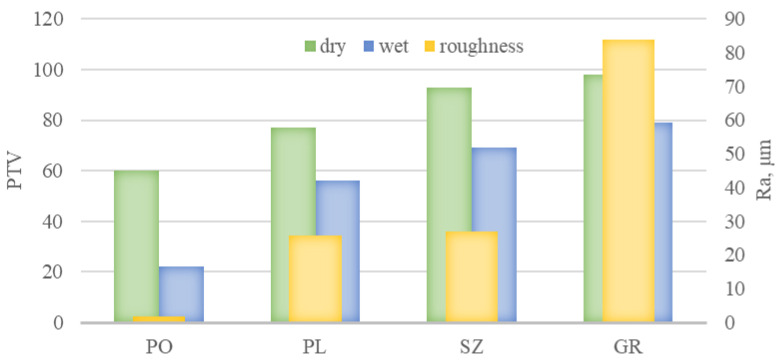
Results of pendulum test value (PTV) slip resistance tests for PO, PL, SZ, and grained (GR) granite slabs under dry and wet conditions against surface roughness expressed as *Ra.*

**Figure 9 materials-14-01108-f009:**
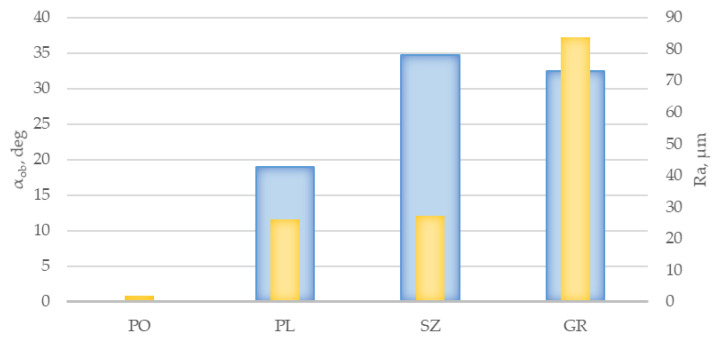
Test results of the acceptance angle (α_ob_) with the shoe foot method for PO, PL, SZ, and GR granite slabs against surface roughness expressed by *Ra*.

**Figure 10 materials-14-01108-f010:**
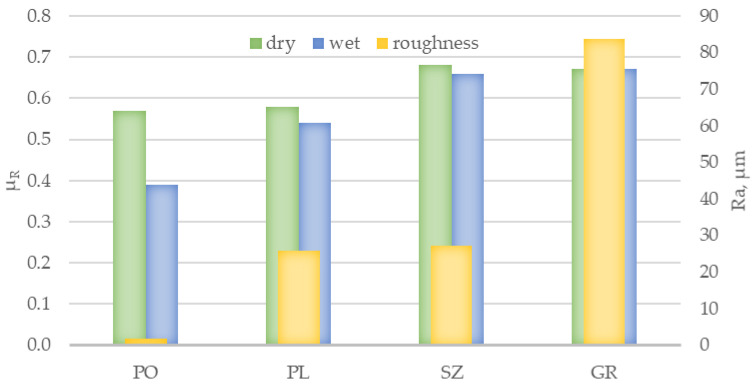
Results of dynamic friction coefficient (μ_R_) of PO, PL, SZ, and GR granite slabs, determined in dry and wet conditions using a set of rubber sliders against the background of surface roughness expressed by *Ra*.

**Figure 11 materials-14-01108-f011:**
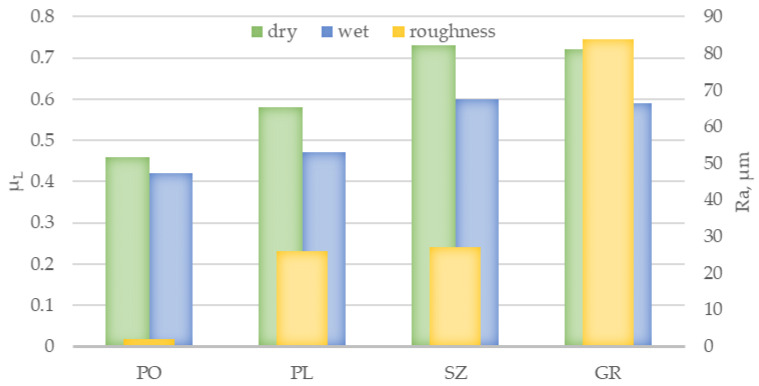
Results of dynamic friction coefficient (μ_L_) of PO, PL, SZ, and GR granite slabs, determined in dry and wet conditions using a set of leather sliders against the background of surface roughness expressed by *Ra*.

**Figure 12 materials-14-01108-f012:**
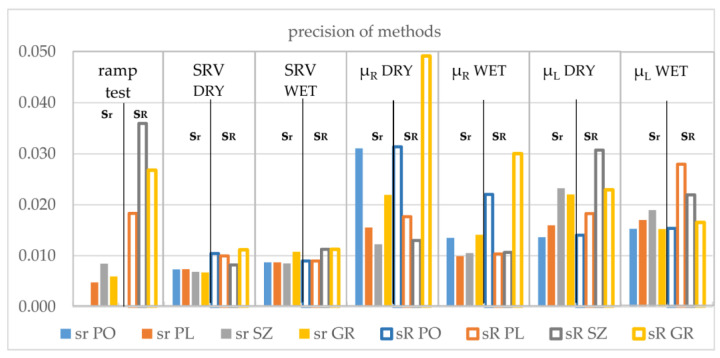
Precision of methods: repeatability standard deviation (s_r_) and reproducibility standard deviation (s_R_) for ramp test, test of SRV for dry and wet surfaces, sliding friction coefficient using rubber slider (µ_R_), and sliding friction coefficient using leather slider (µ_L_) for dry and wet surfaces. The values of s_r_ and s_R_ were obtained from the results subjected to quotient transformations (maximal values for each test method were used as normalizing values).

**Figure 13 materials-14-01108-f013:**
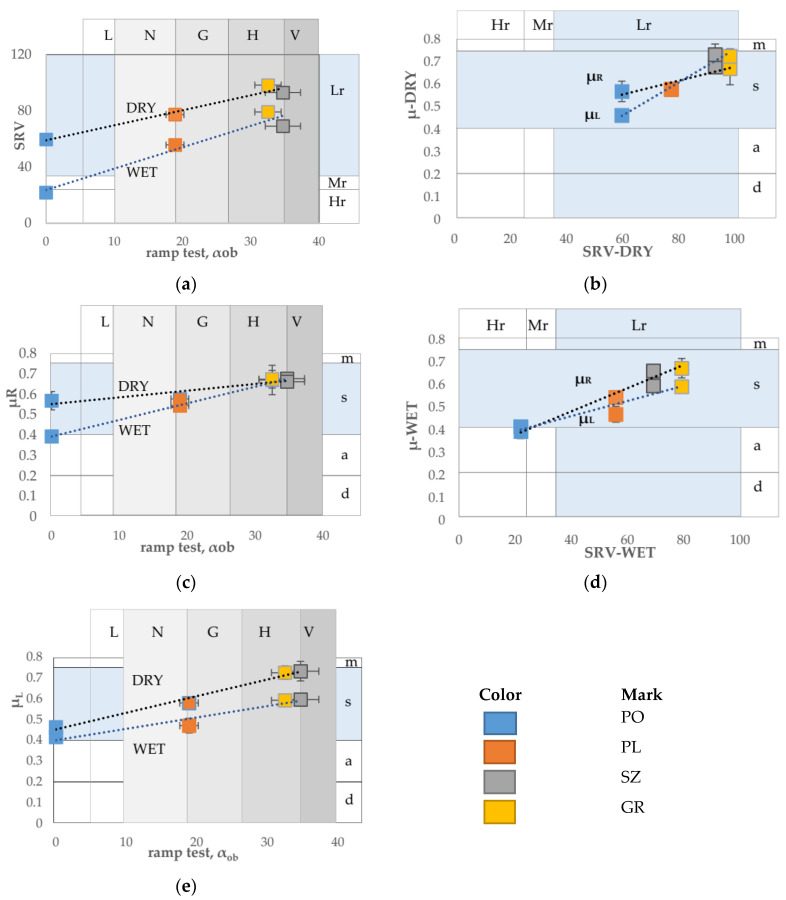
Slip evaluation for four surfaces (PO, PL, SZ, and GR) depending on the test method and associated classification criteria. (**a**) SRV for dry and wet surfaces versus α_ob_ results of the ramp test. (**b**) Sliding friction coefficient using rubber slider (µ_R_) for dry and wet surfaces versus results of the ramp test. (**c**) Sliding friction coefficient using leather slider (µ_L_) for dry and wet surfaces versus results of the ramp test. (**d**) Dry surface µ_R_ and µ_L_ versus SRV. (**e**) Wet surface µ_R_ and µ_L_ versus SRV.

**Table 1 materials-14-01108-t001:** Characteristics of granite slabs.

Sample’s Mark	Processing Texture	Roughness Parameters, μm				Surface Characteristics
		S_a_	S_z_	R_a_	R_z_	
**PO**	polishing	4.05	213.87	1.85	25.96	high degree of smoothness, shine
**PL**	flaming	37.28	149.19	25.93	130.47	appearance close to natural fracture, with clear changes in the surface of the quartz grains caused by temperature and flame
**SZ**	brushed	38.20	556.38	27.11	294.39	clear roughness with abrasive scratches
**GR**	graining	97.11	839.01	83.75	418.39	even but rough surface with characteristic regular concaves and convexities

**Table 2 materials-14-01108-t002:** Pearson’s correlation coefficient between test methods.

Test Mark	SRV (dry)	SRV (wet)	µ_R_ (dry)	µ_R_ (wet)	µ_L_ (dry)	µ_L_ (wet)
Ramp test	0.977	0.970	0.837 *	0.988	0.979	0.955
SRV (dry)	-	0.984	0.863 *	0.990	0.978	0.963
SRV (wet)	-	-	0.783 *	0.981	0.949	0.914 *
µ_R_ (dry)	-	-	-	0.852 *	0.879 *	0.904 *
µ_R_ (wet)	-	-	-	-	0.978	0.953
µ_L_ (dry)	-	-	-	-	-	0.974

If we consider that the results were obtained for four levels (PO, PL, SZ, and GR), the critical value of Pearson’s coefficient for *n* = 4 is much higher and values marked * are smaller than the critical value, but in nonstatistical evaluation such values are usually considered as good correlation.

**Table 3 materials-14-01108-t003:** Values of the intergroup deviation (s_L_)_._ The values of s_L_ for which s_L_^2^ had not met the condition described by Equation (2) are marked with gray shading.

Test Mark	s_L_ (PO)	s_L_ (PL)	s_L_ (SZ)	s_L_ (GR)
Ramp test	0.000	0.018	0.035	0.026
SRV (dry)	0.007	0.007	0.004	0.009
SRV (wet)	0.002	0.002	0.007	0.003
µ_R_ (dry)	0.005	0.008	0.004	0.044
µ_R_ (wet)	0.017	0.003	0.001	0.027
µ_L_ (dry)	0.003	0.009	0.020	0.006
µ_L_ (wet)	0.002	0.022	0.011	0.006

## Data Availability

The data presented in this study are available on request from the corresponding author.

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
