# Peer review of "Comparative Analysis of Slip Resistance Test Methods for Granite Floors"

_materials, 2021, doi:10.3390/ma14051108_

Round 1

Reviewer 1 Report

The paper is an interesting assessment for improving safety on constructions, with a subject worth of publishing. However, I must recommend a few corrections to be made prior to publication:

  1. Please rewrite the title to a more fluid sentence with no period in the middle of it. For instance, “Comparative analysis of slip resistance test methods for granite floors”.
  2. The examples of usage of granite slabs on known buildings in subsection 2.1 would be more appropriately placed in the Introduction section, as section 2 must be more objectively focused on the materials.
  3. Throughout the text and some figures, there are several typos and English and punctuation errors (example: Figure 2 - “high” instead of “hight”; Figure 4 - “admissible” instead of “admissiblen”). Please conduct a thorough proof-reading before resubmitting the paper.
  4. The ramp test described in subsection 2.3 seems too prone to subjectivity. Could you please further describe calibration liner procedure that supposedly eliminates this factor?
  5. Understanding that the assessment of slip resistance is highly dependent on the test method, and that none of them perform well simultaneously in terms of accuracy and resolution, I agree that standardization is needed for this topic. However, the reader that searches for your paper is most likely looking for an answer to the question: “which method should I use then to ensure my particular slab to be as safe as possible against slippage?”. Therefore, I recommend you write a few sentences on which is the most indicated test method given the present state-of-the-art and why, even though it still may have its limitations.

Reviewer 2 Report

Comments on the manuscript:  materials-1112807-peer-review-v1  

 “Slip Resistance of Granite Floors. Comparative 2 Analysis of Test Methods”

General comments:

The manuscript contains an interesting research on the slip friction of granite floors. The authors compared the experimental results by using three methods and four types of granite slabs with roughness between Ra =1.85 microns to Ra =83.75 microns.

A lot of experimental results suggest the importance of the granite slabs roughness  on the friction coefficient, both in dry and in the presence of the water. The obtained results indicate that the test method and classification method have a strong influence  on the slip risk assessment. Also the authors concluded that the results of testing and classification indicate an important need to standardize testing  methods and classification methods  for slip friction risk of various granite floors .

Observations:

  1. It is clear that the surface roughness is an important parameter to determine the mechanical component of the friction coefficient for a sliding tribosistem. For two surfaces having high differences between hardness ( like granite slabs and rubber or leather) the contact pressure influence the friction coefficient. The reviewer suggest to authors to indicate the contact pressure in their experiments. We consider that the tests realized both in “Ramp tests” and in “Sliding friction coefficient” have different contact pressure.

  1. In the “Ramp tests” the authors realized the test by “barefoot method”. Usually people drive with shoes. The friction between foot skin and granite is deferent that between rubber and granite.

  1. In the “Sliding friction coefficient” method the authors must indicate the sliding speeds for all the tests.  Depending on the sliding speed can developed a hydrodynamic effect reducing the coefficient of friction.

  1. In Fig. 9 it can be observe that for polished granite (PO) the angle of acceptance alpha (ob) is zero. That means a very low friction coefficient. But in Fig. 11 the dynamic friction coefficient for the some granite (PO) and leather the friction coefficient is  (0.4-0.45). Or between  angle of acceptance and static friction coefficient can be use following equation: COF = atan (alpha).

Also, if it compare the friction coefficients obtained by equation COF = atan (alpha), where the alpha are the angles presented in Fig. 9  with the friction coefficients presented in Fig. 11, important differences can be obtained.

  1. Some minor corrections must be realized as follows: Fig 9 must be indicate alphaob , (deg) and not (o), in Fig. 10 and 11 please use the some notation for friction coefficient in graphics and in text ( in text is miug!).
